# 4-Substituted Thieno[3,2-*d*]pyrimidines as Dual-Stage Antiplasmodial Derivatives

**DOI:** 10.3390/ph15070820

**Published:** 2022-07-01

**Authors:** Prisca Lagardère, Romain Mustière, Nadia Amanzougaghene, Sébastien Hutter, Jean-François Franetich, Nadine Azas, Patrice Vanelle, Pierre Verhaeghe, Nicolas Primas, Dominique Mazier, Nicolas Masurier, Vincent Lisowski

**Affiliations:** 1Institut des Biomolécules Max Mousseron, UMR 5247, CNRS, Université de Montpellier, ENSCM, UFR des Sciences Pharmaceutiques et Biologiques, 34293 Montpellier, France; prisca.lagardere@umontpellier.fr; 2Aix Marseille Univ, CNRS, ICR UMR 7273, Equipe Pharmaco-Chimie Radicalaire, Faculté de Pharmacie, 13385 Marseille, France; romain.mustiere@etu.univ-amu.fr (R.M.); patrice.vanelle@univ-amu.fr (P.V.); nicolas.primas@univ-amu.fr (N.P.); 3Centre d’Immunologie et des Maladies Infectieuses (CIMI), INSERM, CNRS, Sorbonne Université, 75013 Paris, France; amanzougaghene_nadia@yahoo.fr (N.A.); jean-francois.franetich@upmc.fr (J.-F.F.); dominique.mazier@sorbonne-universite.fr (D.M.); 4Aix Marseille Univ, IRD, AP-HM, SSA, VITROME, 13005 Marseille, France; sebastien.hutter@univ-amu.fr (S.H.); nadine.azas@univ-amu.fr (N.A.); 5Service Central de la Qualité et de l’Information Pharmaceutiques, AP-HM, Hôpital Conception, 13005 Marseille, France; 6CHU de Nîmes, Service de Pharmacie, 30029 Nîmes, France; pierre.verhaeghe@lcc-toulouse.fr; 7LCC-CNRS, Université de Toulouse, CNRS UPR 8241, UPS, 31400 Toulouse, France

**Keywords:** Malaria, thienopyrimidine, erythocytic, hepatic, *Plasmodium*

## Abstract

Malaria remains one of the major health problems worldwide. The increasing resistance of *Plasmodium* to approved antimalarial drugs requires the development of novel antiplasmodial agents that can effectively prevent and/or treat this disease. Based on the structure of Gamhepathiopine, a 2-*tert*-butylaminothieno[3,2-*d*]pyrimidin-4(3*H*)-one hit, active on the sexual and asexual stages of the parasite and thanked for the introduction of various substituents at position 4 of the thienopyrimidine core by nucleophilic aromatic substitution and pallado-catalyzed coupling reactions, a series of 4-substituted thieno[3,2-*d*]pyrimidines were identified as displaying in vitro activities against both the erythrocytic stage of *P. falciparum* and the hepatic stage of *P. berghei.* Among the 28 compounds evaluated, the chloro analogue of Gamhepathiopine showed good activity against the erythrocytic stage of *P. falciparum,* moderate toxicity on HepG2, and better activity against hepatic *P. berghei* parasites, compared to Gamhepathiopine.

## 1. Introduction

Malaria is a dreaded disease, mainly located in the African region [1]. According to the WHO (World Health Organization), this disease infected in 2021 nearly 241 million people worldwide, and killed 627,000 people [2,3]. Despite the prevention policy carried out by the authorities, such as vector control, and the use of preventive antimalarial drugs and the vaccination of children, populations living in regions with moderate to high transmission remain vulnerable to malaria [4,5,6]. To fight against this scourge, the therapeutic arsenal currently proposes the use of artemisinin or its derivatives in combination with other partner drugs in artemisinin-based combination therapies (ACTs) [7,8,9]. ACTs are recommended as the first line of defense against severe *Plasmodium falciparum* malaria, but their effectiveness is threatened by the steady growth of new resistant strains [10,11,12]. Since 2015, a new malaria control program has been developed by the WHO, with the aim of reducing the mortality rate by 90% by 2030 [13]. To achieve this goal, innovative treatments need to be developed, especially with multistage activities against *Plasmodium* [14,15]. In this context, our group identified a thienopyrimidine derivative, named Gamhepathiopine (Figure 1), which presented promising antiplasmodial activity on all the different developmental stages of *P. falciparum* [16,17,18]. A first structure-activity relationship (SAR) study was performed, including the modulation of positions 2 and 6 of the thienopyrimidine scaffold [18]. A *tert*-butylamine at position 2, as well as a *p*-tolyl group at position 6, were found to be the appropriate substituents to maintain the antiplasmodial activity in this series (Figure 1). To further these studies, we report herein the synthesis of new 4-substituted thieno[3,2-*d*]pyrimidine derivatives and their in vitro activities against the blood stage of *P. falciparum* and the liver stage of a rodent malaria model, *P. berghei.* To bring chemical diversity in position 4 of the thienopyrimidine core and to study the influence of a heteroatom at this position, we introduced *O*-alkyl, *O*-aryl, *S*-aryl, (alkyl)amino, and hydrazinyl substituents, as well as an alkynyl or (hetero)aryl group. All these derivatives could be prepared from a common halogenated intermediate, using aromatic nucleophilic substitutions (*O*-, *S*-, and *N*-alkyl or *N*-aryl groups) or Sonogashira and Suzuki–Miyaura coupling reactions (alkyne or (hetero)aryl groups).

## 2. Results and Discussion

### 2.1. Chemistry

Compound **1** was prepared according to the method described by Cohen et al. (Figure 1) [18]. Briefly, 4-methylacetophenone was treated with phosphoryl trichloride and hydroxylamine hydrochloride in dimethylformamide (DMF) at room temperature to generate compound **2** in an 89% isolated yield. This latter was reacted with methyl thioglycolate in the presence of sodium methylate in methanol to afford methyl 3-amino-5-(*p*-tolyl)thiophene-2-carboxylate **3**, which was isolated in an 82% yield. Then, the condensation of **3** with ethoxycarbonyl isothiocyanate in DMF led to a thiourea intermediate, which was then reacted with *tert*-butylamine, triethylamine, and EDCI·HCl (1.2 eq) as a coupling agent to form a guanidine intermediate. Finally, the heating of the reaction mixture allowed its cyclization and subsequent deprotection to offer compound **1** in an overall three-step yield of 85%. The chlorination of **1** led to **5**, which was isolated in a quantitative yield. A halogen exchange was then realized using an excess of sodium iodide in a dioxane reflux. Under these conditions, the conversion was revealed to be slow and incomplete, even after 72 h of reaction. Finally, compound **6** was isolated only in a 12% yield as a formic acid salt, after a preparative HPLC.

The introduction of chemical diversity was then studied from the 4-chloro derivative **5**, as the iodo analog was obtained only in small quantities. First, *O*-alkylation was studied using sodium ethylate or methanolate at 60 °C, leading to compounds **7a**–**b**, which were isolated after recrystallization, in 86 and 88% yields, respectively (Figure 2). The introduction of an *O*-aryl group or an *S*-aryl was then carried out by nucleophilic substitution with (thio)phenolic compounds, in the presence of potassium carbonate as a base. Whereas substitution with phenol derivatives required high temperature to proceed, substitution with thiophenol derivatives was efficient at room temperature. Compounds **8a**–**b** and **9a**–**b** were isolated after purification by chromatography in 54–88% yields.

*N*-Alkylation was then studied by using compound **5** and a set of aliphatic amines or hydrazines (Figure 3). Linear, ramified, or functionalized primary amines were introduced at position 4 of the thienopyrimidine core in the presence of a base in ethanol reflux. Compounds **10c**–**i** were isolated in moderate to good yields (51–77%). In the case of a volatile reagent such as ammonia or (methyl)hydrazine, the reaction was carried out in a sealed tube. Under these conditions, compounds **10a**, **10b**, and **10i** were obtained in high yields after chromatographic purification. Surprisingly, the ^1^H NMR spectrum of **10i** revealed that the N-3 of the pyrimidine ring was protonated, contrary to what was observed with other amines, where the pyrimidine ring retained its aromaticity (Appendix A). Finally, an amino acid (alanine) was introduced using the same strategy. The resulting compound was revealed to be insoluble in DMSO, which did not allow its biological evaluation. To increase its solubility, it was converted into its sodium salt **10j**, using sodium ethylate in ethanol.

The introduction of phenyl groups at position 4 was then studied using a Suzuki–Miyaura reaction (Figure 4). First, the conditions described by Fernandez-Mato et al. were tested [19], using compound **5** and 4-methylphenylboronic acid as starting materials. In the presence of 1.3 equivalents of anhydrous potassium carbonate and 2.5% of Pd(PPh_3_)_4_ in anhydrous toluene at 100 °C for 12 h, only traces of the coupling product were observed in the LC-MS analysis (Table 1, Entry 1). The use of polar solvents such as DMF or THF led to the same results (Entry 2, 3). Then, the experimental conditions described by Oh et al. to synthesize substituted benzothienopyrimidines were tested [20]. Using a 1M aqueous solution of potassium carbonate and 10% of Pd(PPh_3_)_4_ in THF reflux increased the conversion rate of the coupling reaction, and compound **11a** was isolated in a 34% yield after chromatography (Entry 4). Finally, increasing the quantity of the base and catalyst led to the full conversion of the starting materials, and compound **11a** was isolated in a 67% yield (Entry 5).

Using these conditions, a series of five other boronic acids were used and the corresponding 4-(hetero)arylthienopyrimidines **11b**–**f** were obtained in good yields (66–80%) after purification. Three 4-alkynyl-substituted derivatives were synthesized according to the conditions described by Wang et al. [21]. Compounds **12a**–**c** were obtained after purification in low to moderate yields (7–35%), due to purification difficulties (compounds **12a** and **12c**). Finally, the 4-unsubstituted derivative **13** was synthesized from **5**, using the conditions described by Muraoka et al. (Figure 5) [22]. After 3 days of reaction at room temperature, compound **5** was fully converted into **13**, which was isolated after chromatography in a 71% yield. The structure of compound **13** was confirmed by ^1^H NMR with the appearance of a new singlet at 8.88 ppm (Appendix A).

### 2.2. In Vitro Antiplasmodial Activity and Cytotoxicity

All synthesized 4-(un)substituted thienopyrimidines were screened for their activity against drug-resistant *P. falciparum* K1 asexual intra-erythrocytic-stage parasites, and were compared to Gamhepathiopine **1**. This strain is resistant to chloroquine, sulfadoxine, and pyrimethamine. The results are summarized in Table 2.

From compound **1**, which possesses a carbonyl group at position 4, various modulations were studied for this position. The introduction of a chlorine or an iodine atom was tolerated. Compounds **5** and **6** showed similar activities to reference **1** on *Pf.* K1, with an EC_50_ in the submicromolar range. These results indicate that the presence of a hydrogen bond acceptor at this position is not crucial. However, the presence of a substituent remains essential, as the removal of the carbonyl group led to an inactive compound (**13** vs. **1**). Furthermore, the steric hindrance of the iodine atom suggests that this position could tolerate bulky substituents. The introduction of various substituents was then studied. The introduction of an aryl group led to a loss of activity when this group was linked directly to the thienopyrimidine core (compounds **11**), or through an ether (compounds **8**) or a thioether link (compounds **9**). In the case of an amine link, some activity was maintained (compound **10h**), suggesting that the presence of an amino group at this position could be tolerated. Thus, several amino-alkyl derivatives were evaluated. Whereas an amino acid residue (compound **10j**) led to the loss of activity, the presence of a primary amine (compound **10a**), a short linear amino-alkyl (**10c** and **10d**) or a slightly longer chain (**10f** and **10g**) was tolerated at this position and led to compounds with quite similar activities to compound **1** (EC_50_ in the micromolar range). Finally, the introduction of a short ramified amino-alkyl (**10e**) or a hydrazinyl group (compounds **10b** and **10i**) was also tolerated but led to a slightly lower activity than compound **1** (IC_50_ around 2 µM). Finally, a 4-alkynyl substituent at position 4 (compounds **12**) led to totally inactive compounds.

The most active compounds against the erythrocytic stage of *P. falciparum* were then selected for further studies. Seven compounds (namely compounds **5**, **6**, **10a**, **10c**–**d**, and **10f**–**g**), which displayed an EC_50_ lower than 2 µM, were then evaluated for their ability to inhibit hepatic infection by the rodent malaria parasite *P. berghei*. Their cytotoxicity was also evaluated on the human hepatoma cell line HepG2 and on primary simian hepatocytes. The results are summarized in Table 3.

Except compound **10a**, which was not active against *P. berghei*, all selected compounds showed similar or higher activities on the hepatic stage than compound **1**. The best results were obtained with compounds **10c**, **10d**, and **10g**, with EC_50_ around 5 µM, followed by compounds **10f** and **5** with an EC_50_ around 15 µM. These results suggested that the presence of an alkylamine group at position 4 increased activity, with a shorter alkyl group being preferable. Activity decreased with the alkyl chain increasing (EC_50_ of **10c** < EC_50_ of **10d** and **10g** < EC_50_ of **10f**). All compounds displayed a moderate to high toxicity against the HepG2 human cell line, with their CC_50_ values ranging from 2.9 to 29 µM. Surprisingly, these compounds did not show similar toxicity against primary simian hepatocytes, except **10g**, which was cytotoxic. All compounds, except **10f**–**g**, showed a CC_50_ greater than 20 µM. Finally, compound **5** with a chlorine atom at position 4 showed the best compromise, with an EC_50_ against *P. berghei* slightly lower than that of compound **1**, with moderate toxicity on HepG2 (CC_50_ = 16.4 ± 5.4 µM) and no toxicity on primary simian hepatocytes (CC_50_ > 100 µM).

## 3. Materials and Methods

### 3.1. Chemistry

#### 3.1.1. Materials and Methods

Commercial reagents and solvents were used without further purification. The synthesized products were purified by chromatography on silica gel (40–63 μm) or by preparative HPLC on an apparatus equipped with a Delta Pack C18 radial compression column (100 mm × 40 mm, 15 μm, 100 Å) at a wavelength of 214 nm, at a flow rate of 28 mL/min with a variable elution gradient from X % A (H_2_O + 0.1 TFA) to X + 30% B (ACN + 0.1% TFA) in 30 min. The purest fractions are pooled, freeze-dried, and lyophilized to give the final compounds. LC/MS analyses were recorded on a Quattro microTM ESI triple quadrupole mass spectrometer (ESI+ electrospray ionization mode) or on a Micromass ZQ spectrometer (ESI+ electrospray ionization mode), coupled to an Alliance HPLC system (Waters, Milford, CT, USA) equipped with a Chromolith High Resolution RP-18e column (25 × 4.6 mm), with the samples being previously separated using a gradient from 100% (H_2_O + 0.1% HCO_2_H) to 100% (ACN + 0.1% HCO_2_H) in 3 min at a flow rate of 3 mL/min and with UV detection at 214 nm. UPLC/MS analyses were recorded with an Acquity H-Class UPLC system, coupled to a Waters SQDetector-2 mass spectrometer (Waters, Milford, CT, USA). Chromatographic separation was carried out under the same conditions as above using a Waters Acquity UPLC BEC C18 column (100 × 2.1 mm, 1.5 μm). High-resolution mass spectrometry (HRMS) analyses were performed with a time-of-flight (TOF) spectrometer coupled to a positive electrospray ionization (ESI) source. The elemental analysis was realized with a Elementar Vario Micro Cube. The NMR spectra were recorded on a Brüker 400, 500, or 600 spectrometer. Chemical shifts δ are expressed in parts per million (ppm) relative to the residual signal of the deuterated solvent used (CDCl_3_, DMSO-d_6_, MeOH-d_3_, ACN-d_3_), and the coupling constants *J* are expressed in Hertz. The multiplicities are designated as singlet (s), broad singlet (bs) doublet (d), doublet of doublet (dd), triplet (t), quadruplet (q), quintuplet (qt), sextuplet (st), or multiplet (m). Compound **1** was synthesized according to the previous reported procedure and its physical characteristics agreed with the published data [18].

#### 3.1.2. Synthesis of *N*-(*Tert*-butyl)-4-chloro-6-(*p*-tolyl)thieno[3,2-*d*]pyrimidin-2-amine **5**

Phosphorus oxychloride (5.5 eq., 8.37 g, 54.59 mmol, 5.1 mL) was slowly added to a cold solution of compound **1** (1 eq, 3.11 g, 9.93 mmol) and *N*,*N*-dimethylaniline (0.7 eq, 842 mg, 6.95 mmol) in CH_3_CN (156 mL) for 2 h at 0 °C. The mixture was then heated to 80–85 °C and was stirred for 18 h. The reaction mixture was cooled to 40 °C and then was quenched into water (400 mL) for 2 h. The precipitate was filtered and washed with water (200 mL). The desired product was dried under vacuum at 50 °C for 48 h to afford **5** as a green/yellow solid (3.26 g, qt). ^1^H NMR (500 MHz, DMSO-d_6_): δ 7.64 (d, 2H, CH_ar_, *J* = 8.0 Hz), 7.37 (s, 1H, H-thiophene), 7.28 (d, 2H, CH_ar_, *J* = 8.0 Hz), 2.36 (s, 3H, CH_3_-tolyl), 1.46 (s, 9H, C(CH_3_)_3_); ^13^C NMR (125 MHz, DMSO-d_6_): δ 158.8, 157.1, 151.9, 149.9, 138.4, 130.0, 129.2, 125.3, 119.1, 111.0, 50.7, 28.3, 20.2; HR-MS (ESI) calculated for C_17_H_18_ClN_3_S: 332.0988 [M + H]^+^, found: 332.0988.

#### 3.1.3. Synthesis of *N*-(*Tert*-butyl)-4-iodo-6-(*p*-tolyl)thieno[3,2-*d*]pyrimidin-2-amine **6**

Under argon atmosphere, compound **5** (100 mg, 0.3013 mmol) was dissolved in anhydrous dioxane (7 mL/mmol) and sodium iodide was added (6 eq., 338 mg, 1.808 mmol, 0.09 mL). The reaction mixture was stirred under reflux for 72 h. The solvent was removed under reduced pressure. The crude was dissolved in sodium thiosulfate solution (10%, 25 mL). The aqueous layer was washed three times with EtOAc. Organic layers were combined, washed with water and brine, dried with MgSO_4_, and filtered and concentrated under vacuum. The crude was purified by preparative HPLC (ACN/H_2_O + 0.1% HCO_2_H) to afford the desired compound as yellow/brown solid (17 mg, 12%); ^1^H NMR (500 MHz, CDCl_3_): δ 7.60 (d, 2H, CH_ar_, *J* = 8.0 Hz), 7.50 (s, 1H, H-thiophene), 7.26 (d, 2H, CH_ar_, *J* = 8.0 Hz), 5.22 (bs, 1H, NH), 2.40 (s, 3H, CH_3_-tolyl), 1.47 (s, 9H, C(CH_3_)_3_).; ^13^C NMR (125 MHz, DMSO-d_6_): δ 160.3, 158.6, 154.1, 140.4, 130.6, 130.0 (2C), 128.9, 126.5 (2C), 122.1, 119.2, 51.4, 28.9, 21.5; LC/MS: t_r_ = 2.71 min, [M + H]^+^ = 424.0; HR-MS (ESI) calculated for C_17_H_18_IN_3_S: 424.0339 [M + H]^+^, found: 424.0352.

#### 3.1.4. Synthesis of 4-Arylether-thieno[3,2-*d*]pyrimidines **7**

Sodium (2.5 eq.) was dissolved in the desired alcohol (10 mL/mmol) under argon at 0 °C. Then, compound **5** (1 eq.) was added slowly and the reaction mixture was stirred at 60 °C under argon. After completion of the reaction, the medium was evaporated to dryness. A total of 50 mL of water was added and then extracted with diethyl ether (3 × 25 mL). The organic layer was washed with brine, dried with MgSO_4_, and filtered and concentrated under reduced pressure. The crude was recrystallized from ACN.

*N*-(*tert*-butyl)-4-methoxy-6-(*p*-tolyl)thieno[3,2-*d*]pyrimidin-2-amine (**7a**)

Yellow crystals (354 mg, 88%); ^1^H NMR (500 MHz, DMSO-d_6_): δ 7.71 (d, 2H, CH_ar_, *J* = 8.0 Hz), 7.53 (s, 1H, H-thiophene), 7.28 (d, 2H, CH_ar_, *J* = 8.0 Hz), 6.57 (bs, 1H, NH), 4.01 (s, 3H, CH_3_O), 2.34 (s, 3H, CH_3_-tolyl), 1.43 (s, 9H, C(CH_3_)_3_); ^13^C NMR (125 MHz, DMSO-d_6_): δ 164.2, 163.2, 160.4, 150.4, 139.3, 130.2, 129.8, 126.0, 118.6, 104.6, 53.6, 50.2, 28.9, 20.9; HR-MS (ESI) calculated for C_18_H_22_N_3_OS: 328.1484 [M + H]^+^, found: 328.1485.

*N*-(*tert*-butyl)-4-ethoxy-6-(*p*-tolyl)thieno[3,2-*d*]pyrimidin-2-amine (**7b**)

Yellow crystals (231 mg, 86%); ^1^H NMR (500 MHz, DMSO-d_6_): δ 7.72 (d, 2H, CH_ar_, *J* = 8.0 Hz), 7.52 (s, 1H, H-thiophene), 7.28 (d, 2H, CH_ar_, *J* = 8.0 Hz), 6.54 (bs, 1H, NH), 4.49 (q, 2H, CH_2_, *J* = 7.1 Hz), 2.34 (s, 3H, CH_3_-tolyl), 1.43 (s, 9H, C(CH_3_)_3_), 1.38 (t, 3H, CH_3_, *J* = 7.1 Hz); ^13^C NMR (125 MHz, DMSO-d_6_): δ 164.2, 162.8, 160.5, 150.3, 139.3, 130.2, 129.8, 126.0, 118.5, 104.6, 62.0, 50.2, 28.9, 20.9, 14.5; HR-MS (ESI) calculated for C_19_H_24_N_3_OS: 342.1635 [M + H]^+^, found: 342.1638.

#### 3.1.5. Synthesis of 4-Arylether-thieno[3,2-*d*]pyrimidines **8**

The appropriate phenol (1 eq., 0.753 mmol) and K_2_CO_3_ (1.1 eq., 0829 mmol) were dissolved in DMF (5 mL) and the reaction mixture was stirred for 15 min. Then, compound **5** (250 mg, 0.753 mmol) was added and the reaction mixture was heated at 130 °C for 3 h. Once the reaction finished, the reaction mixture returned to room temperature and was poured in water. The aqueous layer was extracted four times with EtOAc. The organic layers were combined, washed with brine, dried with MgSO_4_, and filtered and concentrated under vacuo. The crude product was purified by chromatography on silica gel (eluent: 9/1 Hexane/EtOAc then 8/2) and was recrystallized in ACN.

*N*-(*tert*-butyl)-4-phenoxy-6-(*p*-tolyl)thieno[3,2-*d*]pyrimidin-2-amine (**8a**)

Yellow crystals; (212 mg, 72%); ^1^H NMR (500 MHz, DMSO-d_6_): δ 7.76 (d, 2H, CH_ar_, *J* = 8.2 Hz), 7.59 (s, 1H, H-thiophene), 7.48–7.45 (m, 2H, CH_ar_), 7.32–7.28 (m, 5H, CH_ar_), 6.59 (bs, 1H, NH), 2.36 (s, 3H, CH_3_-tolyl), 1.18 (s, 9H, C(CH_3_)_3_); ^13^C NMR (125 MHz, DMSO-d_6_): δ 165.5, 163.0, 160.4, 152.2, 151.5, 139.6, 130.0, 129.9, 129.6, 126.2, 125.7, 122.2, 118.4, 104.3, 50.2, 28.6, 20.9; HR-MS (ESI) calculated for C_23_H_24_N_3_OS: 390.1635 [M + H]^+^, found: 390.1644.

*N*-(*tert*-butyl)-6-(*p*-tolyl)-4-(*p*-tolyloxy)thieno[3,2-*d*]pyrimidin-2-amine (**8b**)

Yellow crystals (192 mg, 63%); ^1^H NMR (500 MHz, DMSO-d_6_): δ 7.75 (d, 2H, CH_ar_, *J* = 8.0 Hz), 7.59 (s, 1H, H-thiophene), 7.31 (d, 2H, CH_ar_, *J* = 8.0 Hz), 7.25 (d, 2H, CH_ar_, *J* = 8.4 Hz), 7.25 (d, 2H, CH_ar_, *J* = 8.4 Hz), 6.52 (bs, 1H, NH), 2.36 (s, 3H, CH_3_-tolyl), 2.33 (s, 3H, CH_3_-tolyl), 1.22 (s, 9H, C(CH_3_)_3_); ^13^C NMR (125 MHz, DMSO-d_6_): δ 165.4, 163.1, 160.3, 151.4, 149.9, 139.6, 134.7, 130.1, 129.9, 129.8, 126.1, 121.9, 118.5, 104.4, 50.2, 28.6, 20.9, 20.5; HR-MS (ESI) calculated for C_24_H_26_N_3_OS: 404.1791 [M + H]^+^, found: 404.1796.

#### 3.1.6. Synthesis of 4-Thioether-thieno[3,2-*d*]pyrimidines **9**

Compound **5** was dissolved in DMF (5mL/mmol). Then, the appropriate thiophenol (10 eq.) and K_2_CO_3_ (1 eq.) were added. The reaction mixture was stirred at room temperature for 1 to 1h 30. Once the reaction was completed, the reaction mixture was concentrated under reduced pressure and water was added. The aqueous layer was extracted three times with DCM. Organic layers were washed with brine, dried with MgSO_4_, and filtered and concentrated under vacuo. The crude was purified by chromatography on silica gel (eluent: 8/2 Hexane/EtOAc for compound **9a** and 9/1 Hexane/EtOAc then 7/3 for compound **9b**). The isolated product was recrystallized in ACN.

*N*-(*tert*-butyl)-6-(*p*-tolyl)-4-(phenylthio)thieno[3,2-*d*]pyrimidin-2-amine (**9a**)

Yellow crystals (365 mg, 85%); ^1^H NMR (500 MHz, DMSO-d_6_): δ 7.72 (d, 2H, CH_ar_, *J* = 7.9 Hz), 7.68–7.68 (m, 2H, CH_ar_), 7.55–7.48 (m, 4H, CH_ar_ + H-thiophene), 7.30 (d, 2H, CH_ar_, *J* = 8.2 Hz), 6.61 (bs, 1H, NH), 2.35 (s, 3H, CH_3_-tolyl), 1.11 (s, 9H, C(CH_3_)_3_); ^13^C NMR (125 MHz, DMSO-d_6_): δ 161.7, 160.6, 158.9, 150.3, 138.9, 135.1, 129.0, 128.9, 128.8, 128.6, 125.8, 125.3, 117.1, 114.2, 49.0, 27.5, 20.0; HR-MS (ESI) calculated for C_23_H_24_N_3_S_2_: 406.1406 [M + H]^+^, found: 406.1398.

*N*-(*tert*-butyl)-6-(*p*-tolyl)-4-(*p*-tolylthio)thieno[3,2-*d*]pyrimidin-2-amine (**9b**)

Yellow crystals (137 mg, 54%); ^1^H NMR (500 MHz, DMSO-d_6_): δ 7.71 (d, 2H, CH_ar_, *J* = 7.9 Hz), 7.55–7.53 (m, 3H, CH_ar_ + H-thiophene), 7.32–7.29 (m, 4H, CH_ar_), 6.58 (bs, 1H, NH), 2.37 (s, 3H, CH_3_-tolyl), 2.35 (s, 3H, CH_3_-tolyl), 1.12 (s, 9H, C(CH_3_)_3_); ^13^C NMR (125 MHz, DMSO-d_6_): δ 162.5, 162.0, 159.9, 151.3, 139.9, 139.8, 136.1, 130.2, 129.9, 126.3, 123.3, 118.1, 115.2, 50.1, 28.5, 21.0, 20.9; HR-MS (ESI) calculated for C_24_H_26_N_3_S_2_: 420.1568 [M + H]^+^, found: 420.1572.

#### 3.1.7. Synthesis of *N*^2^-(*Tert*-butyl)-6-(*p*-tolyl)thieno[3,2-*d*]pyrimidine-2,4-diamine **10a**

A mixture of **5** (0.5 g, 1.507 mmol) in EtOH/NH_4_OH (35%) in a 1:1 ratio (50 mL) was heated at 120 °C. The reaction mixture was stirred for 14 h in a sealed tube. After completion, the reaction mixture was poured into 50 mL of water and was extracted three times with EtOAc. The combined organic layers were washed three times with brine, dried with MgSO_4_, and filtered and concentrated under reduced pressure. The crude compound was purified by chromatography on silica gel (eluent: 9/1 to 5/5 Hexane/EtOAc) to afford **10a** (295 mg, 63%) as an off-white solid. ^1^H NMR (500 MHz, DMSO-d_6_): δ 7,66 (d, 2H, CH_ar_, *J* = 8.0 Hz), 7,37 (s, 1H, H-thiophene), 7,27 (d, 2H, CH_ar_, *J* = 8.0 Hz), 6,76 (bs, 2H, NH_2_), 5,64 (bs, 1H, NH), 2.34 (s, 3H, CH_3_-tolyl), 1.40 (s, 9H, C(CH_3_)_3_); ^13^C NMR (125 MHz, DMSO-d_6_): δ 162.4, 160.9, 157.6, 147.8, 138.7, 130.7, 129.7, 125.7, 119.0, 109.3, 49.8, 29.1, 20.8; HR-MS (ESI) calculated for C_17_H_21_N_4_S: 313.1487 [M + H]^+^, found: 313.1504.

#### 3.1.8. Synthesis of 4-Hydrazino-thieno[3,2-*d*]pyrimidines **10b** and **10i**

A mixture of compound **5** (1.0 eq.) and the appropriate hydrazine (10 eq.) was stirred and heated under EtOH reflux (3 mL/mmol) in a sealed tube. The reaction mixture was stirred for 4 to 12 h. After completion, the reaction mixture returned to room temperature and was concentrated under reduced pressure. The crude was dissolved in EtOAc and the organic layer was washed three times with brine. The organic layer was dried with MgSO_4_ and filtered and concentrated under reduced pressure. The crude was purified by recrystallization in ACN.

*N*-(*tert*-butyl)-4-(1-methylhydrazinyl)-6-(*p*-tolyl)-3,4-dihydrothieno[3,2-*d*]pyrimidin-2-amine (**10b**)

Brown solid (209 mg, 90%); ^1^H NMR (500 MHz, DMSO-d_6_): δ 7.65 (d, 2H, CH_ar_, *J* = 8.1 Hz), 7.26 (s, 1H, H-thiophene), 7.25 (d, 2H, CH_ar_, *J* = 8.1 Hz), 5.76 (s, 1H, NH), 5.09 (bs, 2H, NH_2_), 3.31 (bs, 3H, CH_3_N), 2.33 (s, 3H, CH_3_-tolyl), 1.40 (s, 9H, C(CH_3_)_3_); ^13^C NMR (125 MHz, DMSO-d_6_): δ 162.3, 160.0, 159.4, 150.2, 138.3, 131.2, 129.7, 125.7, 117.9, 104.0, 49.8, 38.9, 29.2, 20.9; HR-MS (ESI) calculated for C_18_H_24_N_5_S: 342.1747 [M + H]^+^, found: 342.1756.

*N*-(*tert*-butyl)-4-hydrazino-6-(*p*-tolyl)thieno[3,2-*d*]pyrimidin-2-amine (**10i**)

Brown solid (94 mg, 92%); ^1^H NMR (500 MHz, DMSO-d_6_): δ 8.30 (bs, 1H, NH), 7.66 (d, 2H, CH_ar_, *J* = 8.1 Hz), 7.30 (s, 1H, H-thiophene), 7.26 (d, 2H, CH_ar_, *J* = 8.0 Hz), 5.62 (bs, 1H, NH), 4.67 (bs, 2H, NH_2_), 2.33 (s, 3H, CH_3_-tolyl), 1.39 (s, 9H, C(CH_3_)_3_); ^13^C NMR (125 MHz, DMSO-d_6_): δ 162.6, 160.4, 160.3, 150.3, 138.4, 131.1, 129.7, 125.7, 118.1, 102.5, 49.8, 30.8, 29.3, 20.9; HR-MS (ESI) calculated for C_17_H_22_N_5_S: 328.1596 [M + H]^+^, found: 328.1599.

#### 3.1.9. General Procedure for the Synthesis of 4-Amino-thieno[3,2-*d*]pyrimidines **10c**–**h**

Compound **5** (1 eq.) was dissolved in EtOH (22 mL/mmol). The appropriate amine (3 eq.) and Na_2_CO_3_ (2 eq.) were added to the solution. The reaction mixture was heated at 35 °C for compound **10c** or under reflux for other amines. If the reaction was not completed after 72 h, another portion of the appropriate amine was added, and the heating was continued until completion of the reaction (1 eq. was added for compound **10h**; 15 eq. for compound **10c**; and 4.5 eq. for other compounds). Then, the mixture returned to room temperature and was concentrated under reduced pressure. The mixture was dissolved in water. The aqueous layer was extracted three times with EtOAc. Organic layers were washed with brine, dried with MgSO_4_, and filtered and concentrated under vacuo. The crude was purified by the appropriate method.

*N*^2^-(*tert*-butyl)-*N*^4^-methyl-6-(*p*-tolyl)thieno[3,2-*d*]pyrimidine-2,4-diamine (**10c**)

Following the general procedure, using **5** (237 mg, 0.742 mmol) and methylamine. The crude was purified by chromatography on silica gel (eluent: 5/5 Hexane/AcOEt) to afford **10c** as a yellow solid (175 mg, 75%); ^1^H NMR (500 MHz, DMSO-d_6_): δ 7.63 (d, 2H, CH_ar_, *J* = 8.1 Hz), 7.29 (s, 1H, H-thiophene), 7.28 (d, 2H, CH_ar_, *J* = 7.9 Hz), 5.76 (bs, 1H, NH), 3.31 (s, 3H, CH_3_N), 2.33 (s, 3H, CH_3_-tolyl), 1.40 (s, 9H, C(CH_3_)_3_); ^13^C NMR (125 MHz, DMSO-d_6_): δ 161.3, 160.4, 156.8, 147.0, 138.1, 130.4, 129.2, 125.4, 118.6, 104.4, 49.5, 28.9, 26.9, 20.2; HR-MS (ESI) calculated for C_18_H_22_N_4_S: 327.1643 [M + H]^+^, found: 4327.1643.

*N*^2^-(*tert*-butyl)-*N*^4^-propyl-6-(*p*-tolyl)thieno[3,2-*d*]pyrimidine-2,4-diamine (**10d**)

Following the general procedure, using **5** (300 mg, 0.904 mmol) and propylamine. The crude compound was purified by chromatography on silica gel (eluent: 9/1 to 7/3 Hexane/EtOAc) to afford **10d** as a yellow solid (214 mg, 67%); ^1^H NMR (400 MHz, DMSO-d_6_): δ 7.65 (d, 2H, CH_ar_, *J* = 8.0 Hz), 7.33–7.31 (m, 2H, H-thiophène + NHCH_2_), 7.28 (d, 2H, CH_ar_, *J* = 8.0 Hz), 5.79 (bs, 1H, NH), 3.41–3.36 (m, 2H, CH_2_N), 2.34 (s, 3H, CH_3_-tolyl), 1.65–1.56 (m, 2H, CH_2_), 1.40 (s, 9H, C(CH_3_)_3_), 0.91 (t, 3H, CH_3_CH_2_, *J* = 7.4 Hz); ^13^C NMR (400 MHz, DMSO-d_6_): δ 161.8, 160.7, 155.8, 147.2, 138.6, 130.7, 129.7, 125.7, 118.8, 104.2, 49.8, 41.5, 29.2, 22.4, 20.8; HR-MS (ESI) calculated for C_20_H_27_N_4_S: 355.1956 [M + H]^+^, found: 355.1960.

*N*^2^-(*tert*-butyl)-*N*^4^-isopropyl-6-(*p*-tolyl)thieno[3,2-*d*]pyrimidine-2,4-diamine (**10e**)

Following the general procedure, using **5** (300 mg, 0.904 mmol) and isopropylamine. The crude was purified by a manual chromatographic column of silica gel (eluent: 9/1 Hexane/AcOEt to 6/4). Compound **10e** was obtained as a white solid (238 mg, 74%); ^1^H NMR (600 MHz, DMSO-d_6_): δ 7.64 (d, 2H, CH_ar_, *J* = 8.0 Hz), 7.32 (s, 1H, H-thiophene), 7.28 (d, 2H, CH_ar_, *J* = 8.0 Hz), 7.06 (d, 1H, NHCH, *J* = 7.6 Hz), 5.76 (bs, 1H, NH), 4.42–4.34 (m, 1H, CHN), 2.34 (s, 3H, CH_3_-tolyl), 1.40 (s, 9H, C(CH_3_)_3_), 1.21 (d, 6H, C(CH_3_)_2_, *J* = 6.5 Hz); ^13^C NMR (150 MHz, DMSO-d_6_): δ 161.8, 160.7, 155.8, 147.2, 138.6, 130.7, 129.7, 125.7, 118.8, 104.2, 49.8, 41.5, 29.2, 22.4, 20.8; HR-MS (ESI) calculated for C_20_H_27_N_4_S: 355.1956 [M + H]^+^, found: 355.1956.

4-((2-(*tert*-butylamino)-6-(*p*-tolyl)thieno[3,2-*d*]pyrimidin-4-yl)amino)butan-1-ol (**10f**)

Following the general procedure, using **5** (150 mg, 0.45.2 mmol) and 4-amino-1-butanol. The crude compound was purified by recrystallization to afford **10f** as pale yellow crystals (134 mg, 77%); ^1^H NMR (500 MHz, DMSO-d_6_): δ 7.65 (d, 2H, CH_ar_, *J* = 8.0 Hz), 7.36–7.34 (m, 2H, H-thiophene + NHCH_2_), 7.27 (d, 2H, CH_ar_, *J* = 8.0 Hz), 5.82 (bs, 1H, NH), 4.45 (t, 1H, OH, *J* = 5.0 Hz), 3.45–3.40 (m, 4H, CH_2_N + CH_2_O), 2.34 (s, 3H, CH_3_-tolyl), 1.64–1.58 (m, 2H, CH_2_), 1.52–1.46 (m, 2H, CH_2_), 1.40 (s, 9H, C(CH_3_)_3_); ^13^C NMR (125 MHz, DMSO-d_6_): δ 161.7, 160.7, 156.5, 147.3, 138.7, 130.7, 129.8, 125.7, 118.9, 104.2, 60.7, 49.9, 40.1, 30.2, 29.3, 25.9, 20.9; HR-MS (ESI) calculated for C_21_H_29_N_4_OS: 385.2062 [M + H]^+^, found: 385.2068.

*N*^2^-(*tert*-butyl)-*N*^4^-(2-methoxyethyl)-6-(*p*-tolyl)thieno[3,2-*d*]pyrimidine-2,4-diamine (**10g**)

Following the general procedure, using **5** (150 mg, 0.45.2 mmol) and 2-methoxyethylamine. The crude compound was purified by a silica gel chromatography (eluent: 8/2 to 6/4 Hexane/EtOAc) to afford **10g** as a yellowish solid (127 mg, 76%); ^1^H NMR (500 MHz, DMSO-d_6_): δ 7.65 (d, 2H, CH_ar_, *J* = 8.2 Hz), 7.40 (t, 1H, NHCH_2_, *J* = 4.7 Hz), 7.35 (s, 1H, H-thiophene), 7.28 (d, 2H, CH_ar_, *J* = 8.2 Hz), 5.88 (bs, 1H, NH), 3.61–3.58 (m, 2H, CH_2_N), 3.53–3.50 (m, 2H, CH_2_O), 3.27 (s, 3H, CH_3_O), 2.34 (s, 3H, CH_3_-tolyl), 1.40 (s, 9H, C(CH_3_)_3_); ^13^C NMR (125 MHz, DMSO-d_6_): δ 161.9, 160.6, 156.5, 147.5, 138.7, 130.7, 129.8, 125.8, 118.9, 104.2, 70.6, 58.0, 49.9, 39.76, 29.2, 20.9; HR-MS (ESI) calculated for C_20_H_27_N_4_OS: 371.1906 [M + H]^+^, found: 371.1904.

*N*^2^-(*tert*-butyl)-*N*^4^-phenyl-6-(*p*-tolyl)thieno[3,2-*d*]pyrimidine-2,4-diamine (**10h**)

Following the general procedure, using **5** (250 mg, 0.753 mmol) and aniline. The crude was then purified by chromatography on silica gel (eluent: 9/1 Hexane/AcOEt to 8/2). Compound **10h** was obtained as a brown solid (149 mg, 51%); ^1^H NMR (500 MHz, DMSO-d_6_): δ 9.01 (bs, 1H, NH), 7.83 (d, 2H, CH_ar_, *J* = 7.5 Hz), 7.70 (d, 2H, CH_ar_, *J* = 8.2 Hz), 7.47 (s, 1H, H-thiophene), 7.34–7.29 (m, 4H, CH_ar_), 7.05–7.03 (m, 1H, CH_ar_), 6.61 (bs, 1H, NH), 2.35 (s, 3H, CH_3_-tolyl), 1.40 (s, 9H, C(CH_3_)_3_); ^13^C NMR (125 MHz, DMSO-d_6_): δ 162.8, 160.2, 154.6, 148.5, 139.8, 139.0, 130.5, 129.9, 128.4, 125.8, 122.6, 121.5, 119.1, 105.0, 50.0, 29.0, 20.9; LC/MS: t_r_ = 1.68 min, [M + H]^+^ = 389.2; HR-MS (ESI) calculated for C_23_H_25_N_4_S: 389.1794 [M + H]^+^, found: 389.1811.

#### 3.1.10. Synthesis of Sodium (*S*)-2-((2-(*Tert*-butylamino)-6-(*p*-tolyl)thieno[3,2-*d*]pyrimidin-4-yl) amino)-propanoate trihydrate **10j**

Compound **5** (200 mg, 0.603 mmol) was dissolved in EtOH (17 mL/mmol). Alanine (3 eq., 161 mg, 1.808 mmol) and Na_2_CO_3_ (2 eq., 192 mg, 1.808 mmol) were added to the solution. The reaction mixture was heated under reflux for 8 days. Then, the mixture returned to room temperature and was concentrated under reduce pressure. Water (50 mL) was added, and the precipitate was filtered and dried under vacuum at 50 °C overnight. The desired compound was obtained without further purification as a white powder (129 mg, 56%); ^1^H NMR (500 MHz, DMSO-d_6_): δ 7.64 (d, 2H, CH_ar_, *J* = 8.1 Hz), 7.31–7.28 (m, 3H, H-thiophene + CH_ar_), 7.08–7.07 (m, 1H, NHCH), 5.43 (bs, 1H, NH), 4.66–4.61 (m, 1H, CH), 2.37 (s, 3H, CH_3_-tolyl), 1.47 (d, 3H, CH_3_, *J* = 7.2 Hz), 1.42 (s, 9H, C(CH_3_)_3_); ^13^C NMR (125 MHz, DMSO-d_6_): δ 173.7, 161.7, 160.1, 155.7, 147.5, 138.2, 130.4, 129.2, 125.4, 118.5, 104.3, 49.5, 48.6, 28.9, 20.2, 17.2; UPLC/MS: t_r_ = 2.58 min, [M + H]^+^ = 385.3; HR-MS (ESI) C_20_H_25_N_4_O_2_S calculated: 385.1698 [M + H]^+^, found: 385.1700. This compound (79 mg, 0.2055 mmol) was dissolved in anhydrous EtOH. Sodium ethoxide, 21% w/w in ethanol (1.1 eq., 73 mg, 0.226 mmol) was added and the mixture was stirred at room temperature for 7h. The volatiles were removed under reduce pressure. The desired product was obtained as a yellow powder (82 mg, 87%); ^1^H NMR (400 MHz, DMSO-d_6_): δ 7.68 (d, 2H, CH_ar_, *J* = 8.0 Hz), 7.35 (s, 1H, H-thiophene), 7.27 (d, 2H, CH_ar_, *J* = 8.0 Hz), 6.65 (d, 1H, NHCH, *J* = 5.1 Hz), 5.77 (bs, 1H, NH), 4.08–4.01 (m, 1H, CH), 2.34 (s, 3H, CH_3_-tolyl), 1.41 (s, 9H, C(CH_3_)_3_), 1.36 (d, 3H, *J* = 7.2 Hz); ^13^C NMR (100 MHz, DMSO-d_6_): δ 174.1, 161.6, 161.0, 155.4, 146.8, 138.7, 130.5, 129.7, 125.8, 119.1, 104.1, 50.9, 49.8, 29.3, 20.8, 19.7.LC/MS: t_r_ = 1.41 min, [M + H]^+^ = 385.1; HR-MS (ESI) C_20_H_25_N_4_O_2_S calculated: 385.1693 [M + H]^+^, found: 385.1695.Elemental analysis calculated for the trihydrate salt: C_20_H_29_N_4_NaO_5_S: C, 52.16; H, 6.35; N, 12.17; S, 6.96; found: C, 52.07; H, 5.99; N, 11.82; S, 6.83.

#### 3.1.11. General Procedure for the Synthesis of 4-Aryl-thieno[3,2-*d*]pyrimidines **11a**–**f**

Compound **5** (1 eq.) was dissolved in THF (15 mL/mmol) (for compound **12a**–**d**) or 1,4-dioxane (for compounds **12e**–**f**). The mixture was degassed, filled with argon. Then, the appropriate boronic acid (2 eq.) was added and the mixture was heated (30 to 40 °C) to solubilize the starting material. A solution of 1 M potassium carbonate in water (2.6 eq.) and Pd(PPh_3_)_4_ (0.02 eq.) were added. The reaction mixture was stirred and heated under reflux until completion of the reaction. The resulting mixture was partitioned between DCM and water. The organic layer was washed with a saturated aqueous solution of sodium bicarbonate, dried with MgSO_4_, and filtered and concentrated under vacuum. The residue was purified by column chromatography.

*N*-(*tert*-butyl)-4,6-di-*p*-tolylthieno[3,2-*d*]pyrimidin-2-amine (**11a**)

Following the general procedure, starting from **5** (300 mg, 0.904 mmol) and 4-tolylboronic acid (246 mg, 1.808 mmol). The crude was purified by column chromatography (eluent: 98/02 Cyclohexane/EtOAc to 80/20). A yellow solid was collected (236 mg, 67%); ^1^H NMR (500 MHz, DMSO-d_6_): δ 8.02 (d, 2H, CH_ar_, *J* = 8.1 Hz), 7.78 (d, 2H, CH_ar_, *J* = 8.2 Hz), 7.66 (s, 1H, H-thiophene), 7.42 (d, 2H, CH_ar_, *J* = 8.0 Hz), 7.30 (d, 2H, CH_ar_, *J* = 7.9 Hz), 6.73 (bs, 1H, NH), 2.41 (s, 3H, CH_3_-tolyl), 2.35 (s, 3H, CH_3_-tolyl), 1.47 (s, 9H, C(CH_3_)_3_); ^13^C NMR (125 MHz, DMSO-d_6_): δ 164.4, 160.9, 158.5, 152.1, 140.6, 139.8, 134.7, 129.9, 129.5, 127.8, 126.2, 118.7, 115.5, 50.3, 28.8, 21.1, 20.9; LC/MS: t_r_ = 2.65 min, [M + H]^+^ = 388.4; HR-MS (ESI) calculated for C_24_H_26_N_3_S: 388.1842 [M + H]^+^, found: 388.1835.

*N*-(*tert*-butyl)-4-(*m*-tolyl)-6-(*p*-tolyl)thieno[3,2-*d*]pyrimidin-2-amine (**11b**)

Following the general procedure, starting from **5** (300 mg, 0.904 mmol) and 3-tolylboronic acid (246 mg, 1.808 mmol). The crude was purified by column chromatography (eluent: 80/20 Hexane/EtOAc to 60/40) to offer a yellow solid (252 mg, 72%); ^1^H NMR (500 MHz, DMSO-d_6_): δ 7.90–7.88 (m, 2H, CH_ar_), 7.80 (d, 2H, CH_ar_, *J* = 8.0 Hz), 7.68 (s, 1H, H-thiophene), 7.51 (t, 1H, CH_ar_, *J* = 7.6 Hz), 7.41 (d, 1H, CH_ar_, *J* = 7.6 Hz), 7.31 (d, 2H, CH_ar_, *J* = 8.0 Hz), 6.77 (bs, 1H, NH), 2.44 (s, 3H, CH_3_-tolyl), 2.36 (s, 3H, CH_3_-tolyl), 1.47 (s, 9H, C(CH_3_)_3_); ^13^C NMR (125 MHz, DMSO-d_6_): δ 164.4, 160.9, 158.8, 152.3, 138.3, 137.5, 131.3, 129.9, 128.9, 128.4, 126.3, 125.0, 118.7, 115.8, 50.3, 28.8, 21.2, 20.9; LC/MS: t_r_ = 2.68 min, [M + H]^+^ = 388.3; HR-MS (ESI) calculated for C_24_H_26_N_3_S: 388.1842 [M + H]^+^, found: 388.1841.

*N*-(*tert*-butyl)-4-phenyl-6-(*p*-tolyl)thieno[3,2-*d*]pyrimidin-2-amine (**11c**)

Following the general procedure, starting from **5** (300 mg, 0.904 mmol) and phenylboronic acid (246 mg, 1.808 mmol). The crude was purified by column chromatography (eluent: 90/10 Hexane/EtOAc) to offer a yellow solid (268 mg, 79%); ^1^H NMR (500 MHz, DMSO-d_6_): δ 8.12–8.10 (m, 2H, CH_ar_), 7.80 (d, 2H, CH_ar_, *J* = 8.2 Hz), 7.68 (s, 1H, H-thiophene), 7.65–7.58 (m, 3H, CH_ar_), 7.31 (d, 2H, CH_ar_, *J* = 7.9 Hz), 6.80 (bs, 1H, NH), 2.35 (s, 3H, CH_3_-tolyl), 1.48 (s, 9H, C(CH_3_)_3_); ^13^C NMR (125 MHz, DMSO-d_6_): δ 164.5, 160.9, 158.6, 152.4, 139.9, 137.5, 130.7, 129.9, 129.9, 129.0, 127.9, 126.3, 118.7, 115.7, 50.3, 28.8, 20.9; LC/MS: t_r_ = 2.52 min, [M + H]^+^ = 374.1; HR-MS (ESI) calculated for C_23_H_24_N_3_S: 374.1685 [M + H]^+^, found: 374.1680.

*N*-(*tert*-butyl)-4-(thiophen-3-yl)-6-(*p*-tolyl)thieno[3,2-*d*]pyrimidin-2-amine (**11d**)

Following the general procedure, starting from **5** (300 mg, 0.904 mmol) and 3-thiopheneboronic acid (246 mg, 1.808 mmol). The crude was purified by column chromatography (eluent: 90/10 Hexane/EtOAc to 85/15) to offer a brown solid (273 mg, 80%); ^1^H NMR (500 MHz, DMSO-d_6_): δ 8.39–8.38 (m, 1H, H-thiophene), 7.86–7.85 (m, 1H, H-thiophene), 7.81–7.79 (m, 3H, H-thiophene + CH_ar_), 7.66 (s, 1H, H-thiophene), 7.32 (d, 2H, CH_ar_, *J* = 8.0 Hz), 6.70 (bs, 1H, NH), 2.36 (s, 3H, CH_3_-tolyl), 1.47 (s, 9H, C(CH_3_)_3_); ^13^C NMR (125 MHz, DMSO-d_6_): δ 164.4, 160.6, 153.7, 152.0, 139.9, 139.7, 129.9, 128.3, 127.7, 126.9, 126.3, 118.6, 114.9, 50.3, 28.8, 21.0; LC/MS: t_r_ = 2.51 min, [M + H]^+^ = 380.2; HR-MS (ESI) calculated for C_21_H_22_N_3_S_2_: 380.1250 [M + H]^+^, found: 380.1254.

*N*-(*tert*-butyl)-4-(pyridin-4-yl)-6-(*p*-tolyl)thieno[3,2-*d*]pyrimidin-2-amine (**11e**)

Following the general procedure, starting from **5** (177 mg, 0.5345 mmol) and 4-pyridinylboronic acid (146 mg, 1.069 mmol). The crude was purified by column chromatography (eluent: 60/40 Hexane/EtOAc to 100% EtOAc) to offer a yellow solid (133 mg, 66%); ^1^H NMR (500 MHz, CDCl_3_): δ 8.83 (d, 2H, CH_pyridinyl_, *J* = 5.5 Hz), 8.03 (d, 2H, CH_pyridinyl_, *J* = 5.5 Hz), 7.65 (d, 2H, CH_ar_, *J* = 8.2 Hz), 7.43 (s, 1H, H-thiophene), 7.28–7.26 (m, 2H, CH_ar_), 5.37 (bs, 1H, NH), 2.41 (s, 3H, CH_3_-tolyl), 1.56 (s, 9H, C(CH_3_)_3_); ^13^C NMR (125 MHz, CDCl_3_): δ 165.0, 160.8, 157.0, 154.4, 150.7, 145.4, 140.5, 130.5, 130.0, 126.7, 122.4, 118.4, 111.7, 51.4, 29.8, 21.5; LC/MS: t_r_ = 2.47 min, [M + H]^+^ = 375.0; HR-MS (ESI) calculated for C_22_H_23_N_4_S: 375.1638 [M + H]^+^, found: 375.1641.

*N*-(*tert*-butyl)-4-(pyridin-3-yl)-6-(*p*-tolyl)thieno[3,2-*d*]pyrimidin-2-amine (**11f**)

Following the general procedure, starting from **5** (200 mg, 0.6027 mmol) and 3-pyridinylboronic acid (148 mg, 1.205 mmol). The crude was purified by column chromatography (eluent: 70/30 Hexane/EtOAc to 100% EtOAc) to offer a yellow solid (171 mg, 76%); ^1^H NMR (500 MHz, DMSO-d_6_): δ 9.41 (d, 1H, CH_pyridinyl_, *J* = 1.8 Hz), 8.76 (dd, 1H, CH_pyridinyl_, *J* = 4.8 and 1.8 Hz), 8.45–8.43 (m, 1H, CH_pyridinyl_), 7.65 (d, 2H, *J* = 8.2 Hz, CH_ar_), 7.50–7.47 (m, 1H, CH_pyridinyl_), 7.41 (s, 1H, H-thiophene), 7.27–7.26 (m, 2H, CH_ar_), 5.21 (bs, 1H, NH), 2.41 (s, 3H, CH_3_-tolyl), 1.55 (s, 9H, CH_3_-tolyl); ^13^C NMR 125 MHz, DMSO-d_6_): δ 165.2, 161.1, 156.8, 153.9, 151.3, 149.7, 140.3, 135.7, 134.0, 130.6, 130.0, 126.6, 123.7, 118.6, 117.7, 51.2, 29.3, 21.5; LC/MS: t_r_ = 2.41 min, [M + H]^+^ = 375.1; HR-MS (ESI) calculated for C_22_H_23_N_4_S: 375.1638 [M + H]^+^, found: 375.1637.

#### 3.1.12. General Procedure for the Synthesis of 4-Alkynyl-thieno[3,2-*d*]pyrimidines **12a**–**c**

Under nitrogen atmosphere, compound **5**, the appropriate alkyne, copper iodide (16% mol), bis(triphenylphosphine)palladium(II) dichloride (4% mol), and triethylamine (10 eq.) were dissolved in dry ACN (1:1 v:v with triethylamine) in a sealed vial. The obtained suspension was stirred for 10 min under microwave irradiation at 100 °C. Water (20 times the quantity of ACN) was added to the reaction mixture, which was then extracted with dichloromethane. The organic layer was washed with water and the excess was removed under reduced pressure. The obtained crude was purified via the appropriate method.

*N*-*tert*-butyl-6-(4-methylphenyl)-4-(phenylethynyl)thieno[3,2-*d*]pyrimidin-2-amine (**12a**)

Following the general procedure, starting from **5** (0.26 g, 0.78 mmol) and phenylacetylene (103.2 µL, 0.94 mmol). The obtained crude was purified via chromatography (cyclohexane/DCM). Fractions of interest were triturated in *n*-pentane, affording **12a** as a yellow solid (109 mg, 35%); ^1^H NMR (DMSO-d_6_, 400 MHz): δ 7.91 (d, 2H, CH_ar_, *J* = 7.9 Hz), 7.73–7.65 (m, 3H, CH_ar_), 7.60–7.49 (m, 3H, CH_ar_), 7.32 (d, 2H, *J* = 7.9 Hz, CH_ar_), 6.96 (bs, 1H, NH), 2.36 (s, 3H, CH_3_-tolyl), 1.44 (s, 9H, C(CH_3_)_3_); ^13^C NMR (DMSO-d_6_, 100 MHz) δ 163.1, 160.7, 152.8, 143.6, 140.0, 132.1 (2C), 130.5, 129.9 (2C), 129.8, 129.1 (2C), 126.3 (2C), 120.9, 120.2, 118.8, 94.3, 85.3, 50.3, 28.6 (3C), 20.9. HR-MS (ESI) calculated for C_25_H_24_N_3_S: 398.1685 [M + H]^+^, found 398.1678.

*N*-*tert*-butyl-4-(cyclopropylethynyl)-6-(4-methylphenyl)thieno[3,2-*d*]pyrimidin-2-amine (**12b**)

Starting from **5** (0.32 g, 0.96 mmol) and ethynylcyclopropane (409 µL, 4.82 mmol). The obtained crude was purified via flash chromatography (cyclohexane/EtOAc), affording **12b** as a dark brown solid (165 mg, 47% yield); ^1^H NMR (DMSO-d_6_, 400 MHz) δ 7.77 (d, 2H, CH_ar_, *J* = 8.2 Hz), 7.61 (s, 1H, H-thiophene), 7.30 (d, 2H, CH_ar_, *J* = 7.9 Hz), 6.79 (bs, 1H, NH), 2.36 (s, 3H, CH_3_-tolyl), 1.76–1.68 (m, 1H, CH), 1.40 (s, 9H, C(CH_3_)_3_), 1.09–1.01 (m, 2H, CH_2_), 0.91–0.82 (m, 2H, CH_2_); ^13^C NMR (DMSO-d_6_, 100 MHz) δ 162.7, 160.1, 152.4, 144.3, 139.9, 129.9, 129.8 (2C), 126.2 (2C), 118.7, 101.1, 72.4, 50.2, 28.6 (3C), 26.3, 20.9, 9.33 (2C), −0.3. HRMS (ESI) m/z calculated for C_22_H_24_N_3_S [M + H]^+^ 362.1685, found 362.1685.

4-[2-*tert*-butylamino-6-(4-methylphenyl)thieno[3,2-*d*]pyrimidin-4-yl]but-3-yn-1-ol (**12c**)

Starting from **5** (0.32 g, 0.96 mmol) and but-3-yn-1-ol (219 µL, 2.89 mmol), the obtained crude was purified via two successive flash chromatography (DCM/MeOH and then cyclohexane/EtOAc), affording **12c** as a yellow solid (24 mg, 7% yield); ^1^H NMR (DMSO-d_6_, 400 MHz): δ 7.77 (d, 2H, CH_ar_, *J* = 8.2 Hz), 7.62 (s, 1H, H-thiophene), 7.31 (d, 2H, CH_ar_, *J* = 8.2 Hz), 6.81 (bs, 1H, NH), 5.01 (t, 1H, OH, *J* = 5.5 Hz), 3.69–3.61 (m, 2H, CH_2_O), 2.71 (t, 2H, CH_2_, *J* = 6.6 Hz), 2.36 (s, 3H, CH_3_-tolyl), 1.41 (s, 9H, C(CH_3_)_3_); ^13^C NMR (DMSO-d_6_, 100 MHz): δ 162.8, 160.7, 152.6, 144.3, 139.9, 129.9 (3C), 126.3 (2C), 120.8, 118.7, 95.8, 77.9, 59.3, 50.2, 28.6 (3C), 23.2, 20.9; HRMS (ESI) calculated for C_21_H_24_N_3_OS [M + H]^+^ 366.1635, found 366.1636.

#### 3.1.13. Synthesis of *N*-(*Tert*-butyl)-6-(*p*-tolyl)thieno[3,2-*d*]pyrimidin-2-amine **13**

Compound **5** (250 mg, 0.7533 mmol) was dissolved in a solution of ethyl acetate/propan-2-ol, 5/1 (16 mL/mmol). Then, Pd(OH)_2_ (125 mg, 20% on carbon, wetted with ca.50% water) and triethylamine (2.3 eq., 175 mg, 1.733 mmol, 0.24 mL) were added. The suspension was placed under a stream of hydrogen and was stirred at room temperature until total consumption of the starting material (72 h). Three portions of Pd(OH)_2_ were added during the experiment. The reaction mixture was filtered on Celite^®^ and the filtrate was concentrated. The crude was dissolved in EtOAc and washed with a saturated aqueous solution of NaHCO_3_. The aqueous layer was extracted twice with EtOAc. Combined organic layers were washed with brine, dried over MgSO_4_, and filtered and concentrated under reduced pressure. The crude was purified with chromatography of silica (eluent: Hexane/EtOAc 90/10 to 80/20) to afford **14** as an off-white solid (159 mg, 71%); ^1^H NMR (500 MHz, DMSO-d_6_): δ 8.88 (s, 1H, H-pyrimidine), 7.75 (d, 2H, CH_ar_, *J* = 8.2 Hz), 7.60 (s, 1H, H-thiophene), 7.31 (d, 2H, CH_ar_, *J* = 8.0 Hz), 6.70 (bs, 1H, NH), 2.35 (s, 3H, CH_3_-tolyl), 1.42 (s, 9H, C(CH_3_)_3_); ^13^C NMR (125 MHz, DMSO-d_6_): δ 162.4, 160.4, 152.5, 152.2, 139.7, 130.1, 129.9 (2C), 126.2 (2C), 118.9, 118.3, 50.1, 28.7, 20.9; LC/MS: t_r_ = 2.19 min, [M + H]^+^ = 298.3; HR-MS (ESI) C_17_H_20_N_3_S calculated: 298.1372 [M + H]^+^, found: 298.1384.

### 3.2. Biology

#### 3.2.1. Blood-Stage Antiplasmodial Evaluation

A K1 culture-adapted *P. falciparum* strain resistant to chloroquine, pyrimethamine, and proguanil was used in an in vitro culture. It was maintained in a continuous culture, as described previously by Trager and Jensen [23]. Cultures were maintained in fresh A+ human erythrocytes at 2.5% hematocrit in a complete medium (RPMI 1640 with 25 mM HEPES, 25 mM NaHCO_3_, and 10% of A+ human serum) at 37 °C under a reduced O_2_ atmosphere (gas mixture: 10% O_2_, 5% CO_2_, and 85% N_2_). Parasitemia was maintained daily between 1 and 3%. The *P. falciparum* drug susceptibility test was carried out by comparing quantities of DNA in treated and control cultures of parasite in human erythrocytes, according to a SYBR Green I fluorescence-based method [24] using a 96-well fluorescence plate reader. Compounds, previously dissolved in DMSO (final concentration less than 0.5% *v*/*v*), were incubated in a total assay volume of 200 μL (RPMI, 2% hematocrit and 0.4% parasitemia) for 72 h in a humidified atmosphere (10% O_2_ and 5% CO_2_) at 37 °C, in 96-well flat bottom plates. Duplicate assays were performed for each sample. After incubation, plates were frozen at 20 °C for 24 h. Then, the frozen plates were thawed for 1 h at 37 °C. A total of 15 μL of each sample was transferred to 96-well flat-bottom non-sterile black plates (Greiner Bio-One, Kremsmünster, Austria) already containing 15 μL of the SYBR Green I lysis buffer (2X SYBR Green I, 20 mM Tris base pH 7.5, 20 mM EDTA, 0.008% *w*/*v* saponin, 0.08% *w*/*v* Triton X-100). Negative control treated by solvents (DMSO or H_2_O) and positive controls (chloroquine) were added to each set of experiments. Plates were incubated for 15 min at 37 °C and then read on a TECAN Infinite F-200 spectrophotometer with excitation and emission wavelengths at 485 and 535 nm, respectively. The concentrations of compounds required to induce a 50% decrease in parasite growth (EC_50_ K1) were calculated from three independent experiments.

#### 3.2.2. Cytotoxic Evaluation on HepG2 Cell Line

The HepG2 cell line (hepatocarcinoma cell line purchased from ATCC, ref HB-8065) was maintained at 37 °C, 5% CO_2_, with 90% humidity in MEM supplemented with 10% fetal bovine serum, 1% L-glutamine (200 mM), penicillin (100 U/mL), and streptomycin (100 μg/mL) (complete MEM medium). The evaluation of the cytotoxicity of the tested molecules was performed according to the method of Mosmann [25], with slight modifications. Briefly, 5.103 cells in 100 μL of the complete medium were inoculated into each well of a 96-well plate and incubated at 37 °C in a humidified 5% CO_2_. After 24 h incubation, 100 μL of medium with various product concentrations dissolved in DMSO (final concentration less than 0.5% *v*/*v*) was added and the plates were incubated for 72 h at 37 °C. Triplicate assays were performed for each sample. Each plate well was then microscope-examined for detecting possible precipitate formation before the medium was aspirated from the wells. A total of 100 μL of MTT (3-(4,5-dimethyl-2-thiazolyl)-2,5-diphenyl-2H-tetrazolium bromide) solution (0.5 mg/mL in medium without FCS) was then added to each well. Cells were incubated for 2 h at 37 °C. After this time, the MTT solution was removed and DMSO (100 μL) was added to dissolve the resulting blue formazan crystals. Plates were shaken vigorously (700 rpm) for 10 min. The absorbance was measured at 570 nm, with 630 nm as a reference wavelength using a TECAN Infinite F-200 Microplate Reader. DMSO was used as a blank and doxorubicin (purchased from Sigma Aldrich, St. Louis, MO, USA) was used as a positive control. Cell viability was calculated as a percentage of control (cells incubated without compound). The 50% cytotoxic concentration (CC50) was determined from the dose–response curve by using the TableCurve software 2D v.5.0. CC50 values represent the mean value calculated from three independent experiments.

#### 3.2.3. Hepatic-Stage Antiplasmodial Evaluation

##### Parasite Strains and Sporozoite Isolation

*P. berghei* sporozoites constitutively expressing green fluorescent protein (PbGFP, ANKA strain) [26] were obtained by the dissection of salivary glands from infected *Anopheles stephensi* mosquitoes bred and infected in the insectary facilities of UMR-S 1135 (CIMI, Paris, France). Infected salivary glands were removed by hand dissection, crushed in a potter, and filtered through a 40 μm filter for sporozoite isolation (Cell Strainer, BD BioSciences, San Jose, CA, USA). The sporozoites were counted using a disposable plastic microscope slide (KOVA).

##### Primary Hepatocyte Culture

Primary simian hepatocytes were isolated from liver segments collected from healthy *Macaca fascicularis* from CEA, Fontenay aux Roses, France. All hepatocytes were obtained using collagenase perfusion, as previously described [27], and were immediately cryopreserved. One day before infection, the cryopreserved hepatocytes were thawed at 37 °C and were seeded into collagen-coated (5 µg/cm^2^ rat tail collagen I, Invitrogen, Carlsbad, CA, USA) 96-well plate, at a density of 80,000 cells per well. They were cultured at 37 °C in 5% CO_2_ in William’s E medium (Gibco, Life Technologies, Saint Aubin, France) supplemented with 10% of Fetal Bovine Serum FCIII, 5 × 10^–5^ M hydrocortisone hemisuccinate (Serb Laboratories, Paris, France), 5 μg per ml bovine insulin (Sigma Aldrich, St. Louis, MO, USA), 2 mM L-glutamine, and 0.02 U per ml-0.02 μg per ml penicillin-streptomycin (Life Technologies) until infection with sporozoites.

##### In Vitro Infection and Drug Assays

Sporozoites of *P. berghei* were re-suspended in the above complete medium used for the hepatocytes culture. Simian hepatocytes were inoculated in 50 μL of complete media with 20,000 *P. berghei*-sporozoites/well in 96-well plates. The infected culture plates were centrifuged for 10 min at 900× *g* to allow fast parasite sedimentation onto the target cells and were further incubated with the serial dilution of drugs that were prepared in advance. After 3 h of incubation, cultures were washed and further incubated in a fresh medium containing the appropriate drug concentration, which were changed every 24 h during the study period. Cultured hepatocytes were fixed using 4% paraformaldehyde (PFA) for 15 min at room temperature. Host cell and parasite nuclei were labelled with 4′,6-diamidino-2-phenylindole (DAPI). Upon fixation and immunostaining, cell culture plates were analyzed in order to determine the number and size of the parasites using a Cell Insight High Content Screening platform equipped with the Studio HCS software (Thermo Fisher Scientific, Waltham, MA, USA). The parasite size reduction was calculated on the average object area using the total surface area of each selected object (μm^2^). To assess the cell toxicity of drugs for hepatic cultures, fixed plates were further scanned for the DAPI signal representing host nuclei. The analysis was based on counting of total cell nuclei. GraphPad Prism 7 statistical software (GraphPad. Software, San Diego, CA, USA) was used for data analysis and graphing. All values were expressed as means and standard deviations (SD).

## 4. Conclusions

In summary, we have prepared a new library of 28 thienopyrimidines, based on the structure of a first antiplasmodial hit named Gamhepathiopine (compound **1**), identified to be active notably on all stages of *P*. *falciparum*. The introduction of various substituents at position 4 of the thienopyrimidine core was studied using aromatic nucleophilic substitutions or pallado-catalyzed reactions on the common 4-chloro derivative **5**. Among the 28 compounds evaluated, seven with a halogen or an alkylamino group displayed an EC_50_ lower than 2 µM against the erythrocytic stage of *P. falciparum.* These compounds were then selected and evaluated against the hepatic stage of *P. berghei*. Six of the seven tested compounds were identified as displaying a dual-stage antiplasmodial activity and being more active against the hepatic stage of *P. berghei* than compound **1**. Compound **10c** was the most active, but some toxicity on HepG2 cells was also observed with several of these compounds. Thus, compound **5** showed the best compromise, with an EC_50_ of 0.3 µM against the erythrocytic stage of *P. falciparum*, with moderate toxicity on HepG2 (selective index of 54.7) while presenting better activity than compound **1** against hepatic *P. berghei* parasites. Further modulations on this position are currently under investigation to increase the antiplasmodial activity and reduce the cytotoxicity of these compounds.

## Data Availability

Data is contained within the article and Appendix A.

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
