# Peer review of "4-Substituted Thieno[3,2-d]pyrimidines as Dual-Stage Antiplasmodial Derivatives"

_pharmaceuticals, 2022, doi:10.3390/ph15070820_

Round 1

Reviewer 1 Report

The authors created a library of thienopyrimidines using various synthetic strategies and evaluated their blood and hepatic stages. After resolving the following issues, the manuscript can be accepted for publication in Pharmaceuticals:

1. Scheme 1 is perplexing because the authors used R to denote 4-MePh, Cl, and I. To make things clear and uniform, R could be used to denote Cl and I, while R1 could be used to denote 4-MePh, as shown in Scheme 2.

2. The scanned NMR spectra of the synthesised compounds must be included in the supporting information, which I was unable to locate.

Author Response

Comments to the Authors: « The authors created a library of thienopyrimidines using various synthetic strategies and evaluated their blood and hepatic stages. After resolving the following issues, the manuscript can be accepted for publication in Pharmaceuticals: »

 Answer: We thank the reviewer for his comments and are pleased to see that our work was appreciated.

Comments to the Authors: « 1. Scheme 1 is perplexing because the authors used R to denote 4-MePh, Cl, and I. To make things clear and uniform, R could be used to denote Cl and I, while R1 could be used to denote 4-MePh, as shown in Scheme 2. »

Answer : We agree with the reviewer’s comment and Scheme 1 was modified accordingly.

Comments to the Authors: « 2. The scanned NMR spectra of the synthesised compounds must be included in the supporting information, which I was unable to locate. »

Answer : A copy of NMR spectra of all new compounds was included in the supporting information.

Reviewer 2 Report

This paper continues a structure-activity investigation of an antimalarial drug scaffold. A carbonyl group within the central part of the scaffold is replaced with a variety of O-, S-, N- and C-linked groups. The chemistry is elegantly united through the employment of a common chlorine-containing precursor, which is obtainable from the original drug itself. No compounds that are more potent than the original are identified in this work, but some enhancements are observed in terms of dual activity at two different stages of the malaria life-cycle. The experimental methods are generally well described. Overall I believe that this work constitutes a valuable addition to the literature on antimalarial drug development, and should be published.

Minor issues:

·    A structure of gamhepathiopine should be included somewhere in the introduction, along with a schematic of the known SAR

·    In the discussion, compound 10e is described as being similar in potency to 1, while 10b and 10j are described as being less potent than 1. This does not make sense because 10e is less potent than 10b/j

·    In the experimental section, specify the quantity of solvent required per mmol of substrate for all reactions

Author Response

Comments to the Authors: « This paper continues a structure-activity investigation of an antimalarial drug scaffold. A carbonyl group within the central part of the scaffold is replaced with a variety of O-, S-, N- and C-linked groups. The chemistry is elegantly united through the employment of a common chlorine-containing precursor, which is obtainable from the original drug itself. No compounds that are more potent than the original are identified in this work, but some enhancements are observed in terms of dual activity at two different stages of the malaria life-cycle. The experimental methods are generally well described. Overall I believe that this work constitutes a valuable addition to the literature on antimalarial drug development, and should be published. »

 Answer: We thank the reviewer for his comments and are pleased to see that our work was appreciated.

Comments to the Authors: « A structure of gamhepathiopine should be included somewhere in the introduction, along with a schematic of the known SAR »

Answer: A new figure was added showing the structure of gamehepathiopine and the known SAR, see figure 1.

 Comments to the Authors: « In the discussion, compound 10e is described as being similar in potency to 1, while 10b and 10j are described as being less potent than 1. This does not make sense because 10e is less potent than 10b/j »

Answer: We agree with the reviewer’s comment and this part was modified in the manuscript: « Finally, the introduction of a short ramified amino-alkyl (10e) or a hydrazinyl group (compounds 10b and 10j) was also tolerated but led to a slightly lower activity than compound 1 (IC50 around 2 µM). »

 Comments to the Authors: « In the experimental section, specify the quantity of solvent required per mmol of substrate for all reactions »

Answer: the missing volumes or concentrations were added in the experimental section.

Reviewer 3 Report

The article titled 4-Substituted Thieno[3,2-d] pyrimidines as Dual-Stage Antiplasmodial

Derivatives.

after consideration of major comments.

.

1)      Abstract, a) the results of synthesized compounds were missed.

 b) authors should add conclusion obtained from their study.

c) which standard used in this study

2)      Introduction, a) on what bases authors directed to synthesis the target compounds

b) authors should display and predict the SAR for reported compounds so rational their work.

3)      Results and discussion, a) authors did not discussion NMR fund why?

b)      Authors should discuss the results of Antiplasmodial activities in details  

c)       NMR peaks should assign to corresponding Carbon or proton.

Author Response

Comments to the Authors: "The article titled 4-Substituted Thieno[3,2-d] pyrimidines as Dual-Stage Antiplasmodial Derivatives.  after consideration of major comments. 1)      Abstract : a) the results of synthesized compounds were missed. b) authors should add conclusion obtained from their study. c) which standard used in this study"

Answer: The missing informations were added in the abstrat: “Among the 28 compounds evaluated, the chloro analogue of gamehepathiopine displayed a good activity against the erythrocytic stage of P. falciparum, moderate toxicity on HepG2 and a better activity against hepatic P. berghei parasites, compared to gamehepathiopine.”

 Comments to the Authors: « 2)      Introduction, a) on what bases authors directed to synthesis the target compounds. b) authors should display and predict the SAR for reported compounds so rational their work. »

 Answer: As the target of gamehepathiopine is not known at this time, it is not possible for the moment to predict SAR. As indicated in the introduction part of the manuscript, “a first structure-activity relationship (SAR) study was performed, including the modulation of positions 2 and 6 of the thienopyrimidone scaffold [12]. A tert-butylamine at position 2, as well as a p-tolyl group at position 6, were found to be the appropriate substituents to maintain the antiplasmodial activity in this series.” The position 4 of the thienopyrimidine scaffold was not study yet, we proposed therefore to study the influence of various chemical modulations at this position to determine its impact towards the erythrocytic and hepatic stage activities, as well as potential cytotoxicity.

Comments to the Authors: "3)      Results and discussion, a) authors did not discussion NMR fund why?

Answer: All NMR data are presented in the experimental part, signal attribution has been added and copies of all 1H and 13C spectra have been added in a supplementary file. Moreover, we discussed of NMR only if it was usefull to prove the strucutre of the synthesized compound as for compound 10i (“the 1H NMR spectrum of 10i revealed that the N-3 of the pyrimidine ring was protonated, contrary to what was observed with other amines, where the pyrimidine ring retained its aromaticity). We added information in the text concerning compound 13 (“The structure of compound 13 was confirmed by 1H NMR with the appearance of a new singlet at 8.88 ppm.”).

Comments to the Authors: "b) Authors should discuss the results of Antiplasmodial activities in details

Answer: All paragraph 2.2 is dedicated to the discussion of antiplasmodial activities of these new derivatives (pages 6 to 9).

Comments to the Authors: "c) NMR peaks should assign to corresponding Carbon or proton."

Answer : NMR peaks for all protons have been assigned (see experimental part).